# Less Severe Sepsis in Cecal Ligation and Puncture Models with and without Lipopolysaccharide in Mice with Conditional *Ezh2*-Deleted Macrophages (LysM-Cre System)

**DOI:** 10.3390/ijms24108517

**Published:** 2023-05-10

**Authors:** Pornpimol Phuengmaung, Phuriwat Khiewkamrop, Jiradej Makjaroen, Jiraphorn Issara-Amphorn, Atsadang Boonmee, Salisa Benjaskulluecha, Patcharee Ritprajak, Aleksandra Nita-Lazar, Tanapat Palaga, Nattiya Hirankarn, Asada Leelahavanichkul

**Affiliations:** 1Center of Excellence in Translational Research in Inflammation and Immunology (CETRII), Faculty of Medicine, Chulalongkorn University, Bangkok 10330, Thailand; pphuengmaung@gmail.com; 2Department of Microbiology, Faculty of Medicine, Chulalongkorn University, Bangkok 10330, Thailand; 3Center of Excellence in Immunology and Immune-Mediated Diseases, Chulalongkorn University, Bangkok 10330, Thailand; phuriwat.khi@gmail.com (P.K.); tanapat.palaga@gmail.com (T.P.); 4Medical Microbiology, Interdisciplinary and International Program, Graduate School, Chulalongkorn University, Bangkok 10330, Thailand; 5Center of Excellence in Systems Biology, Research Affairs, Faculty of Medicine, Chulalongkorn University, Bangkok 10330, Thailand; jiradejmak@gmail.com; 6Functional Cellular Networks Section, Laboratory of Immune System Biology, National Institute of Allergy and Infectious Diseases, National Institutes of Health, Bethesda, MD 20892, USA; jiraphorn.issara-amphorn@nih.gov (J.I.-A.); nitalazarau@niaid.nih.gov (A.N.-L.); 7Department of Microbiology, Faculty of Science, Chulalongkorn University, Bangkok 10330, Thailand; atsadang88@gmail.com (A.B.); salisafaii@gmail.com (S.B.); 8Research Unit in Integrative Immuno-Microbial Biochemistry and Bioresponsive Nanomaterials, Department of Microbiology, Faculty of Dentistry, Chulalongkorn University, Bangkok 10330, Thailand; p.ritprajak@gmail.com; 9Division of Nephrology, Department of Medicine, Faculty of Medicine, Chulalongkorn University, Bangkok 10330, Thailand

**Keywords:** sepsis, lipopolysaccharide, macrophages, epigenetics, Ezh2

## Abstract

Despite a previous report on less inflammatory responses in mice with an absence of the enhancer of zeste homologue 2 (Ezh2), a histone lysine methyltransferase of epigenetic regulation, using a lipopolysaccharide (LPS) injection model, proteomic analysis and cecal ligation and puncture (CLP), a sepsis model that more resembles human conditions was devised. As such, analysis of cellular and secreted protein (proteome and secretome) after a single LPS activation and LPS tolerance in macrophages from Ezh2 null (Ezh2^flox/flox^; LysM-Cre^cre/−^) mice (Ezh2 null) and the littermate control mice (Ezh2^fl/fl^; LysM-Cre^−/−^) (Ezh2 control) compared with the unstimulated cells from each group indicated fewer activities in Ezh2 null macrophages, especially by the volcano plot analysis. Indeed, supernatant IL-1β and expression of genes in pro-inflammatory M1 macrophage polarization (*IL-1β* and *iNOS*), *TNF-α*, and *NF-κB* (a transcription factor) were lower in Ezh2 null macrophages compared with the control. In LPS tolerance, downregulated *NF-κB* compared with the control was also demonstrated in Ezh2 null cells. In CLP sepsis mice, those with CLP alone and CLP at 2 days after twice receiving LPS injection, representing sepsis and sepsis after endotoxemia, respectively, symptoms were less severe in Ezh2 null mice, as indicated by survival analysis and other biomarkers. However, the Ezh2 inhibitor improved survival only in CLP, but not LPS with CLP. In conclusion, an absence of Ezh2 in macrophages resulted in less severe sepsis, and the use of an Ezh2 inhibitor might be beneficial in sepsis.

## 1. Introduction

Sepsis is a potentially life-threatening condition in response to severe infection regardless of the organismal causes of the infection [1,2,3], which is roughly divided into the hyperinflammation stage and immune exhaustion (immune paralysis) phase [4,5]. Sepsis-induced hyperinflammation is a well-known cause of sepsis mortality, partly through hypercytokinemia-mediated septic shock, especially at an early phase of sepsis [6]. Meanwhile, sepsis-induced immune exhaustion is developed at the same time or shortly after the hyperinflammation, at least in part, due to immune cell death from overwhelming responses against several stimulators from the pathogens and hosts, referred to as pathogen-associated molecular patterns (PAMPs) and damage-associated molecular patterns (DAMPs), respectively [6]. Subsequently, an inadequate response to control the organism during sepsis-induced immune exhaustion results in another episode of secondary infection and another episode of septic shock from the different pathogens [7]. Due to the opposite direction of immune responses in different phases of sepsis, different strategic treatments are necessary. As such, an anti-inflammatory treatment might be beneficial for sepsis-hyperinflammation to attenuate the unnecessary overwhelming immune responses that might be harmful to the host. Meanwhile, an escalation of immune responsiveness during immune exhaustion may be helpful to enhance the microbial control ability of the host to prevent secondary infections [8,9,10,11,12,13,14,15]. Despite the successful decrease in short-term sepsis mortality due to improved supportive care, immune exhaustion-induced secondary infection seems to become more common [16]. Indeed, immune cell apoptosis, myeloid-derived suppressor cells, regulatory T cells, and lipopolysaccharide (LPS) tolerance are mentioned as underlying mechanisms of sepsis-induced immune exhaustion [17,18,19,20]. Among these topics, data on LPS tolerance in sepsis are relatively fewer compared with those on other mechanisms. The presence of LPS, a major molecule of Gram-negative bacteria, in blood circulation during sepsis (endotoxemia) is common due to Gram-negative bacterial infection and/or the translocation of LPS from the intestine into the blood circulation, referred to as “leaky gut”, and is a common cause of endotoxemia [21,22,23]. Because of the highest abundance of Gram-negative bacteria in the gut compared with other organisms, endotoxemia from a leaky gut is mentioned in several conditions with gut barrier defects, including sepsis [1,2,3]. Subsequently, an adaptation to the prolonged LPS stimulations in sepsis may initiate LPS tolerance [24,25]. Among several models of sepsis, cecal ligation and puncture (CLP) is a standard model used for sepsis hyper-inflammation [14,23,26,27,28,29] that more resembles the human condition than a single LPS injection [30]. Meanwhile, sepsis immune exhaustion consists of several models, based on increased susceptibility of the secondary infection [31,32], and the more severe sepsis in CLP surgery after LPS tolerance, using twice-administered LPS injection, compared with CLP without LPS pre-conditioning is previously mentioned [25]. Indeed, the more severe sepsis, especially bacteremia, in CLP after LPS tolerance compared with CLP alone is matched with the common characteristic of the worsened infection in sepsis-induced immune exhaustion compared with the normal immune regulation [33]. Then, CLP and CLP after LPS tolerance were used as models of hyperinflammation and LPS tolerance-associated immune exhaustion with sepsis, respectively.

Interestingly, the tolerance against LPS, especially in monocytes or macrophages, is possibly due to the epigenetic modifications, chromatin remodeling, and interferences in cell energy status [34,35,36], among which epigenetic alteration is the most extensively studied [37,38]. Epigenetics is the phenotypic alterations without the changes in the DNA sequence for the switch “on” and “off” of DNA transcription through (i) the modifications of DNA and/or histone through several enzymes [3,39], and (ii) noncoding RNA (microRNA) [2]. Among all, the methylation at histone 3 lysine 27 (H3K27) is one of the most common epigenetic processes through histone changes for several cell activities, including after LPS activation [40]. In LPS-activated macrophages, the insertion of methyl groups at lysine 27 on histone 3 (H3K27) by histone demethylase is controlled by the polycomb repressor complex group 2 (PCR2), a repressor molecule consisting of several subunits, including Ezh2 (histone-lysine N-methyltransferase-2 or enhancer of zeste homolog) [41,42], to switch off the DNA transcription (reduce cytokine production) through this histone modification [43,44]. Due to the important Ezh2 catalytic activity on PCR2, Ezh2 overexpression enhances PCR2 inhibitory function with an anti-inflammatory effect [45,46] and Ezh2 deletion should enhance pro-inflammatory responses [47]. Indeed, the enhanced pro-inflammatory effect of Ezh2 blockage is mentioned via (i) increased tumoricidal impact of tazemetostat (an Ezh2 inhibitor) [48] and (ii) worsened colitis after *Ezh2* downregulation [49]. In contrast, the anti-inflammatory property of Ezh2 blockage is also mentioned in the atherosclerosis model [50]. Hence, Ezh2 not only downregulates pro-inflammatory cytokines but also can decrease the anti-inflammatory process, partly through the downregulation of the suppressor of cytokine signaling 3 (*Socs3*) [51]. Ezh2 causes histone methylation that block transcription of both pro-inflammatory genes (cytokines) and anti-inflammatory genes (Sosc3) and the impact of Ezh2 might be different among various genes and cell types. Indeed, in mice with conditional Ezh2 deletion by the LysM-Cre system, Ezh2 deletion only in the myeloid cells (monocytes, macrophages, and neutrophils) demonstrates less severe responses against a single LPS injection, but not LPS tolerance (two LPS injections), and bone marrow-derived macrophages from these mice indicated less potent LPS responses (lower supernatant cytokines) and less severe LPS tolerance (higher supernatant cytokines) compared with the control cells [39]. Although Ezh2 impacts on sepsis are still inconclusive, Ezh2 is one of the upregulated genes in LPS-tolerant macrophages [52] which is improved by the Ezh2 inhibitor (enhanced TNF-α expression) [53] as an interesting control of macrophage through epigenetic manipulation [13,54].

Here, the influence of Ezh2 on LPS was further explored through proteomic analysis and tested in a model with sepsis hyper-inflammatory responses (CLP) and a sepsis model after LPS tolerance (twice-administered LPS injection before CLP surgery) using the conditional Ezh2 deletion mice and an Ezh2 inhibitor.

## 2. Results

### 2.1. Proteomic Analysis of Lipopolysaccharide (LPS)-Induced Macrophages from Control and Ezh2 Null Mice

The difference in bone marrow-derived macrophage (BMDM) after activation with three protocols, namely control, a single LPS stimulation, and LPS tolerance (Figure 1A), of macrophages from Ezh2 control (Ezh^fl/fl^; LysM-Cre^−/−^) or Ezh2 null (Ezh^fl/fl^; LysM-Cre^cre/−^) mice were analyzed by proteome and secretome, using the cells and cell supernatant, respectively. With a single LPS stimulation, there were prominent alterations in the peptides of Ezh2 control macrophages compared with the neutral state of Ezh2 control cells as indicated by the up- and downregulation at 188 and 119 proteins, respectively (Figure 1B, left upper). Meanwhile, the values of LPS-activated Ezh2 null cells were 32 and 9 proteins, respectively (Figure 1B, lower left), suggesting possible less activity of Ezh2 null macrophages compared with that in the cells from littermate control mice. Likewise, the up- and downregulated molecules in LPS tolerance of Ezh2 control macrophages compared with the neutral state were 392 and 296 proteins, respectively (Figure 1B, upper right), while for the Ezh2 null cells they were 107 and 68 proteins, respectively (Figure 1B, lower right). With the fold enrichment pathway analysis (ShinyGo 0.77), most of the proteins were correlated to immune response pathways and cell energy status in macrophages from both mouse strains with either single or twice LPS stimulation (Figure 2). The proteins with aerobic respiration and interferon-gamma responses were the groups with the highest fold enrichment in Ezh2 control cells and Ezh2 null macrophages, respectively, after a single LPS activation (Figure 2, left). In LPS tolerance, the proteins with interferon-gamma responses and the negative regulation of innate immune responses were the groups with the highest fold enrichment in Ezh2 control cells and Ezh2 null macrophages, respectively (Figure 2, right). These data implied similar downstream LPS responses in macrophages of both mouse strains. Although there was no direct comparison between Ezh2 control macrophages versus Ezh2 null cells, the Venn diagram analysis, using the Venny 2.1 program (https://bioinfogp.cnb.csic.es/tools/venny) accessed on 15 May 2023, from the list of proteins roughly indicated differences in the deviation of the macrophage proteome away from the resting control condition (activated cells versus control cells) (Figure 3). With a single LPS, 307 and 42 proteins were up- or downregulated in macrophages of littermate control mice (LPS Ezh2 control vs. Ezh2 control) and Ezh2 null mice (LPS Ezh2 null vs. Ezh2 null), respectively, with 32 unique proteins presenting only in the latter group (Figure 3, upper). Because the unique proteins in Ezh2 null macrophages might be responsible for the phenotypic differences between Ezh2 null versus Ezh2 control cells after LPS activation, these proteins were further evaluated by the ShinyGO 0.77 program. Interestingly, these 32 proteins were the member of only 2 pathways, including nuclear factor-κB (NF-κB) and Toll-like receptor (TLR) pathways (Figure 3, upper). Similarly, after LPS tolerance, 1248 and 1269 proteins were altered from the neutral state (up- or downregulation) in the proteome of macrophages from littermate control mice (Ezh2 control) and Ezh2 null mice, respectively, with 104 unique peptides presenting only in Ezh2 null macrophages (Figure 3, lower) which also were mostly associated with cell energy status and immune responses (Figure 3, lower). Details of the proteins from macrophages uniquely elevated in Ezh2 null cells but not in Ezh2 control macrophages are indicated in Appendix A.

As expected, analysis of the secreted proteins from supernatant (secretome analysis) that deviated from the neutral status demonstrated the lessor proteins (Figure 4) compared with the analysis from the cell lysate (Figure 2). There were 6 and 23 up- and downregulated proteins, respectively, in the secretome of LPS Ezh2 control vs. Ezh2 control and 2 and 22 up- and downregulated proteins, respectively, in the secretome of LPS Ezh2 null vs. Ezh2 null (Figure 4, left). Meanwhile, there were 26 and 66 up- and downregulated proteins, respectively, in the secretome of LPS/LPS Ezh2 control vs. Ezh2 control and 8 and 1 up- and downregulated proteins, respectively, in the secretome of LPS/LPS Ezh2 null vs. Ezh2 null (Figure 4, right). Additionally, The Venn diagram demonstrated 106 and 11 overlapped proteins after the activation compared with the neutral state in the cells from each mouse strain after a single LPS and LPS tolerance, respectively (Figure 5). The fold enrichment pathway analysis of the unique proteins in Ezh2 null macrophages indicated an involvement in cell energy status and responses against infection in a single LPS stimulation (Figure 5, upper), while mostly involved in responses to infection in the LPS tolerance group (Figure 5, lower). Details of the proteins in secretome analysis that were uniquely elevated in Ezh2 null cells but not in Ezh2 control macrophages are indicated in Appendix A. These data implied fewer activities of Ezh2 null macrophages compare with the control cells after activation by either LPS or LPS tolerance that might be responsible for the phenotypic responses against LPS.

### 2.2. Less Prominent M1 Macrophage Polarization in LPS-Activated Ezh2 Null Cells with the Downregulation of NF-κB after A Single and Twice LPS Stimulation

Then, macrophages from Ezh2 control (Ezh^fl/fl^; LysM-Cre^−/^) and Ezh2 null (Ezh^fl/fl^; LysM-Cre^cre/−^) mice were tested. Here, the supernatant was removed and the cells were washed 1 day post-incubation before the difference between single and twice LPS stimulations was determined at 2 days to control the duration of culture in both groups (Figure 6A). As such, both a single (N/LPS) and LPS tolerance (LPS/LPS) upregulated M1 macrophage polarization compared with control, as determined by supernatant interleukin (IL)-1β with upregulated *IL-1β* and inducible nitric oxide synthase (*iNOS*), without an alteration in genes of M2 polarization, including resistin-like-α (Retnla or *Fizz-1*), arginase-1 (*Arg-1*), and transforming growth factor beta (*TGF-β*) (Figure 6B–F). In addition, the characteristic of LPS tolerance, a less potent response to the following LPS stimulations compared with the first response to LPS [55,56,57] was demonstrated in both control and *Ezh2* null cells, as the gene expression of *TNF-α* and *IL-6* (but not *IL-10*) in N/LPS were more prominent than LPS tolerance (Figure 6G–I). However, the expression of genes for pro-inflammatory molecule nuclear factor kappa B (*NF-kB*) was similarly higher than in the control group (N/N) (Figure 6J). On the other hand, there was a less prominent M1 polarization (pro-inflammatory macrophages), as indicated by *IL-1β* and *iNOS*, together with less pro-inflammatory responses (*TNF-α*, *IL-6*, and *NF-κB*) in Ezh2 null macrophages compared with control cells after a single LPS stimulation (N/LPS) (Figure 6C–J) supporting a pro-inflammatory effect of Ezh2 on macrophages [39]. In LPS tolerance (LPS/LPS), there was a non-difference in supernatant IL-1β, macrophage polarization, and cytokine genes between Ezh2 null macrophages and control cells, despite lower *NF-κB* expression in Ezh2 null macrophages (Figure 6B–J). Between Ezh2 null macrophages with a single LPS and LPS tolerance, all of these parameters were similar (Figure 6B–J). Despite lower *NF-kB* (a pro-inflammation molecule) in Ezh2 null macrophages with LPS tolerance compared with control cells (Figure 6J), supernatant IL-1β (Figure 6B) and expression of cytokine genes (Figure 6G–I) in Ezh2 null macrophages were similar to that in control cells, implying a limited Ezh2 impact on the control of macrophage responses during LPS tolerance. Although there was a limited impact on LPS tolerance, these data supported an anti-inflammatory response of Ezh2 null macrophages after a single LPS stimulation that might be useful as an anti-inflammation in sepsis.

### 2.3. Characteristics of Ezh2 Control (Ezh^fl/fl^; LysM-Cre^−/−^) or Ezh2 Null (Ezh^fl/fl^; LysM-Cr ^cre/−^) Mice after Cecal Ligation and Puncture (CLP) and LPS Tolerance before CLP Surgery

To investigate the impact of Ezh2 in macrophages on hyper-inflammatory sepsis and sepsis after LPS tolerance, CLP after PBS injection (CLP) and CLP after twice LPS administration (LPS-CLP), respectively, in Ezh2 control (Ezh^fl/fl^; LysM-Cre^−/−^) and Ezh2 null (Ezh^fl/fl^; LysM-Cr ^cre/−^) mice was performed (Figure 7A). In the survival analysis, CLP after LPS tolerance (LPS-CLP) in control mice showed the most severe sepsis as all mice died within 72 h post-surgery (Figure 7A). The more severe sepsis in LPS-CLP mice compared with CLP, especially in the control mice (Figure 7A), implied a possible impact of LPS tolerance on a defect of microbial control. Interestingly, the best survival rate of Ezh2 null mice with CLP and the better survival rate after LPS-CLP of Ezh2 null compared with control mice (Figure 7A) indicated a possible beneficial impact of Ezh2 blockage in macrophages during sepsis. Among the control group, LPS-CLP demonstrated more severe sepsis than CLP alone as indicated by cell-free DNA, bacteremia, and pro-inflammatory cytokines (TNF-α and IL-6), but not other parameters (serum creatinine, alanine transaminase, renal histology score, spleen apoptosis, endotoxemia, and IL-10) (Figure 7C–L and Figure 8). These data indicated more severe sepsis after LPS tolerance compared with no LPS tolerance in the Ezh2 control mice, possibly due to immune exhaustion. However, there was no difference in most of the sepsis severity biomarkers between Ezh2 null mice with LPS-CLP and those with CLP alone, except the higher histology score in LPS-CLP Ezh2 null mice (Figure 7C–L). The data implied no or less immune exhaustion after LPS tolerance in Ezh2 null mice, as mentioned in previous publications [39,58]. Between Ezh2 null versus Ezh2 control mice with CLP-induced sepsis hyperinflammation (CLP) and LPS tolerance with subsequent sepsis (LPS-CLP), sepsis severity was more severe in Ezh2 control mice as indicated by survival analysis, organ injury (kidney and liver), cell-free DNA, endotoxemia, bacteremia, and serum cytokines (TNF-α and IL-6, but not IL-10) (Figure 7C–L), supporting a beneficial impact of Ezh2 deletion in macrophages during sepsis in both conditions.

### 2.4. Ezh2 Inhibitor Attenuated Cecal Ligation and Puncture (CLP) Sepsis in Wild-Type (WT) Mice with Less Impact on CLP after LPS Tolerance

Due to the reduced sepsis severity of Ezh2 null over Ezh2 control mice from our data and the control of inhibitory Socs3 by Ezh2 gene from previous publications [59,60], an Ezh2 inhibitor was further tested, similar to the experiments on Ezh2 null mice mentioned above (Figure 9A). In WT mice, CLP after LPS tolerance (LPS-CLP) also demonstrated the highest mortality rate as all mice died within 96 h of the observation, while approximately 25% of the mice survived at 96 h post-surgery with CLP alone (Figure 9B), supporting more severe sepsis after LPS tolerance similar to Ezh2 control mice (Figure 7A–L). However, Ezh2 inhibitor attenuated disease severity only in CLP alone, but not LPS-CLP (Figure 9B), perhaps because of the more severe sepsis in LPS-CLP compared with CLP alone, as indicated by higher cell-free DNA and serum cytokines (TNF-α and IL-6) in vehicle-administered LPS-CLP mice compared with the CLP alone group (Figure 9E,H,I). In CLP alone, the Ezh2 inhibitor attenuated kidney damage (serum creatinine) and serum cytokines (TNF-α, IL-6, and IL-10) but not cell-free DNA, endotoxemia, and bacteremia (Figure 9B–J). On the other hand, the Ezh2 inhibitor did not attenuate LPS-CLP mice, as indicated by the non-difference in survival analysis, organ damage (kidney and liver), endotoxemia, and bacteremia (Figure 9B–D,F,G), despite the reduction of cell-free DNA and serum cytokines (TNF-α, IL-6, and IL-10) in Ezh2 inhibitor-administered mice (Figure 9E,H–J). 

## 3. Discussion

Lipopolysaccharide (LPS) is one of the potent activators of macrophages, as indicated by the up- and downregulation of several proteins compared with the neutral state of the cells. The macrophages from Ezh2 littermate control or Ezh2 null mice without activation were used as the controls to see the deviation away from the neutral state due to the possible differences between littermate control and Ezh2 null mice. Less profound activities of Ezh2 null macrophages were indicated by a lower number of proteins with similar functions (immune responses, cytokines, and cell energy) between control and Ezh2 null cells, implying similar downstream signals. Because of (i) the possibly reduced adverse effects of selective inhibition only on macrophages but not all cells in the body [61], (ii) endotoxemia in several conditions [62,63,64], partly through gut barrier damage [1,22,24,26] with profound LPS recognition by macrophages [65,66] and (iii) the epigenetic regulation in LPS responses [67], LPS was used to test Ezh2 null macrophages. Notably, Ezh2 reduced chromatin accessibility by adding methyl marks at the tail of histone H3, and the presence of trimethylation of H3K27 (H3K27me3) at promoter regions [68] is mentioned in sepsis [69]. Indeed, the increased mortality in patients with high *Ezh2* and H3K27 is mentioned [70,71] and inhibition of Ezh2 [72] might be beneficial. M1 macrophage polarization (*IL-1β* and *iNOS*) along with pro-inflammatory molecules (*TNF-α*, *IL-6*, and *NF-κB*) were less prominent in LPS-activated Ezh2 null macrophages than the control (Figure 6), possibly correlated with more profound suppressor of cytokine signaling 3 (*Socs3*; an anti-inflammatory molecule) as mentioned in our previous publication [39]. Perhaps *Ezh2* more potently inhibits some molecules than others. In a single LPS stimulation, Ezh2 more profoundly suppressed *Socs3* than *NF-κB* (a transcriptional factor for several cytokines), resulting in less inhibition of *NF-κB* with high cytokine production. Without Ezh2, elevated *Socs3* downregulated *NF-κB* and led to lower *TNF-α* and *IL-6* expression after a single LPS stimulation. Indeed, cytosolic *Socs3* inhibits the NF-κB-dependent inflammatory genes through enhanced ubiquitination and proteasomal degradation [44,59]. Additionally, Ezh2 blockage enhances anti-inflammatory *Socs3* that inhibits hyperinflammation in sepsis (here and others), multiple sclerosis, and glucose-activated peritoneal fibrosis [60,69,73]. In contrast, suppression of Ezh2 might enhance pro-inflammatory genes, including *NF-κB*, that possibly worsens inflammatory bowel diseases and muscle cell apoptosis in sepsis [49,74,75,76]. Although Socs3 can inhibit both anti- and pro-inflammatory cytokines [77], possibly driven by different molecules [78], Socs3 seems to have less impact on anti-inflammatory IL-10 as serum TNF-α and IL-6, but not IL-10, were lower in LPS-injected Ezh2 null mice [39]. Accordingly, Socs3 might be correlated with IL-10 in macrophages responses because (i) IL-10 directly upregulates *Socs3* [79], (ii) Socs3 and IL-10 are simultaneously used to inhibit inflammation [80,81], and (iii) Ezh2 inhibitor (EPZ-6438) upregulated *IL-10* [82].

In mice, the deletion of Ezh2 only in myeloid cells in Ezh2 null mice was enough to induce less severe sepsis, similar to the LPS injection model [39] and pneumococcal sepsis model [70,83], highlighting the major impact of macrophages in the cytokine production [84]. Then, blockage of cytokine production only on the myeloid cells, especially macrophages, might be beneficial. Despite the improved sepsis mortality with the previously known hepatic protection of Ezh2 inhibitor [85], liver enzyme and bacteremia were not different from sepsis in Ezh2 control mice (Figure 7) implying a possible hepatoxicity and incomplete Ezh2 blockage of the selected inhibitor (GSK126). Notably, injection of Ezh2 inhibitor [86] in mice inhibits Ezh2 in all cells which might be harmful in some cell types and the selective Ezh2 blockage only in macrophages might reduce the adverse effect. However, the balance between the pro- and anti-inflammatory directions (a yin–yang effect) after Ezh2 blockage might be different in individual patients.

Because leaky gut causes endotoxemia [1,66] and relatively low inflammation (repeat or chronic exposure of LPS), or LPS tolerance [24,87] possibly causes more severe sepsis [25] and/or secondary infection [31,88,89], CLP surgery after LPS tolerance (LPS-CLP) might be different from CLP without LPS priming (CLP). In control mice, LPS-CLP was more severe than in those with CLP alone, possibly due to inadequate inflammation to control organisms at an early phase of sepsis [25] leading to a higher blood bacterial burden and higher cytokines than in the CLP alone group. Because chronic endotoxemia might induce LPS tolerance [25,57], sepsis in these individuals, such as obesity and uremia, might be more severe than the non-endotoxemia cases [90,91,92,93,94,95], partly, due to LPS tolerance. Despite the higher disease severity of LPS-CLP over CLP alone, Ezh2 null mice still demonstrated less severe sepsis compared with LPS-CLP in control mice, supporting the anti-inflammatory effect of Ezh2 deletion in macrophages. Although the Ezh2 inhibitor did not reduce LPS-CLP mortality, there was less severe systemic inflammation, implying a possible benefit of the dose adjustment. More studies are interesting. Despite the broad spectrum antiviral of Ezh1/2 inhibitors [96], the influence of Ezh1/2 blockage on bacterial infection needs further testing. In cancer, Ezh2 blockage might induce anti-inflammatory macrophages, resulting in less severe sepsis or more infection susceptibility due to the inadequate inflammation to control organisms. Importantly, an effective antibiotic with good microbial control is a main strategy for the treatment of sepsis-induced hyper-inflammation [97]. Hence, our results are a proof of concept to use clinically available Ezh2 inhibitors in sepsis, especially the blockage of Ezh2 specifically only in macrophages.

Several limitations should be mentioned. First, only male mice were used and the impact of gender differences in sepsis might affect the translation of the results. Second, details of mechanistic pathways, including the Western blot analysis, were not performed. Despite a proof of concept for the translational use of Ezh2 inhibitors in clinical sepsis, more experiments would be interesting. Third, the results need to be validated in human situations before a solid conclusion. Nevertheless, we concluded that Ezh2 inhibitors, an available anticancer treatment, might be beneficial in some situations of sepsis. More studies are warranted.

## 4. Materials and Methods

### 4.1. Animal

The Institutional Animal Care and Use Committee of the Faculty of Medicine, Chulalongkorn University, Bangkok, Thailand approved the protocol (No. 017/2562) according to the National Institutes of Health (NIH) criteria. Wild-type (WT) 8-week-old C57BL/6 male mice were purchased from Nomura Siam, Pathumwan, Bangkok, Thailand. Meanwhile, Ezh2^flox/flox^ and LyM-Cre^Cre/Cre^ mice were obtained from RIKEN BRC Experimental Animal Division (Ibaraki, Japan) and cross-bred until having *Ezh2* littermate control (Ezh^fl/fl^; LysM-Cre^−/−^) or *Ezh2* null (Ezh^fl/fl^; LysM-Cre^cre/−^) in F3 generation of the breeding protocol. As such, the Ezh2^flox/flox^ mice had loxP sites upstream and downstream of the 2.7 kb SET domain, while bred with LysM-Cre^Cre/Cre^ mice, the mice with a Cre recombinase under the control of lysozyme M to target *Ezh2* for deletion in myeloid cells (macrophages and neutrophils). Mice with Ezh2^flox/flox^ without LysM-Cre (Ezh^fl/fl^; LysM-Cre^−/−^) were used as littermate controls (Ezh2 control). To genotype these mice on the loxP sites’ insertion, the following primers were used for *Ezh2*: reverse 1: 3′ of loxp: 5′-AGG GCA TCA GCC TGG CTGTA-3′; forward 2: 5′ of loxp: 5′-TTA TTC ATA GAG CCA CCTGG-3′; forward 3: left loxp: 5-ACG AAA CAG CTC CAG ATTCAG GG-3′ according to a previous publication [83]. The mice homozygous for the flox were selected and genotyped for the expression of LysM-Cre using the primers; forward: 5′-CTTGGGCTGCCAGAATTCTC-3′; Reverse: 5′-CCCAGAAATGCCAGATTACG-3′.

### 4.2. Animal Models

Cecal ligation and puncture (CLP) surgery was used to induce sepsis, following previous publications, under isoflurane anesthesia [98,99,100]. Briefly, a median abdominal incision was performed and the cecum was ligated at 10 cm from the cecal tip, punctured twice with a 21-gauge needle, and gently squeezed to express a small amount of fecal material before closing the abdominal wall layer by layer with sutures. Then, tramadol (25 mg/kg/dose) in 0.25 mL prewarmed normal saline solution (NSS) and imipenem/cilastatin (14 mg/kg/dose) in 0.2 mL NSS were subcutaneously administered in abdominal areas after surgery, and at 6 and 18 h post-CLP [9]. In sham-operated mice, the cecum was isolated and closed by suturing without ligation or puncture. In parallel, a sham operation was performed with only cecal identification before closing abdomen layer by layer. Because lipopolysaccharide (LPS) tolerance inhibits macrophage cell respiration and induces global proteomic changes in macrophages [35], sepsis during LPS tolerance might be different from the regular condition. Then, CLP after LPS tolerance using twice-administered LPS injection (LPS-CLP) was conducted. Hence, the mice were divided into 3 groups. First, for CLP in LPS tolerance (LPS-CLP), intraperitoneal injection of 0.8 mg/kg LPS (*Escherichia coli* 026:B6) (Sigma-Aldrich, St. Louis, MO, USA) with another dose of 4 mg/kg LPS at 5 days later and followed by CLP surgery at 1 days after the 2nd dose of LPS was performed. Second, for CLP alone (CLP), the experiments started with intraperitoneal injection of phosphate buffer solution (PBS) at 0 and 5 days followed by CLP surgery. Third, in sham control mice (Sham), 2 doses of PBS at 0 and 5th days of experiments followed by sham surgery was conducted. Of note, the lower 1st dose (0.8 mg/kg) followed by the higher 2nd LPS dose (4 mg/kg) for LPS tolerance induction was performed according to a previous protocol [25]. Mice were sacrificed with cardiac puncture under isoflurane anesthesia with sample collection at 24 h or 96 h post-surgery for blood biomarkers and survival analysis, respectively. On the other hand, these mouse protocols, including CLP, LPS-CLP, and sham, were also used for the test of Ezh2 inhibitor using WT mice in all groups. The Ezh2 inhibitor (GSK343; Medchemexpress, Monmouth, NJ, USA) at 4 mM/25 g mice in 3% dimethyl sulfoxide (DMSO) or DMSO alone (vehicle control) was subcutaneously administered 15 min before surgery and 6 h later (15 min before tramadol and the antibiotics). These mice were sacrificed with the same protocol of experiments in the transgenic mice.

### 4.3. Mouse Sample Analysis

Serum creatinine and alanine transaminase [40] were measured by colorimetric method (QuantiChrom™ Creatinine Assay Kit, BioAssay System, Hayward, CA, USA) and EnzyChrom Alanine Transaminase assay (EALT-100, BioAssay), respectively. Serum cell-free DNA and LPS (endotoxin) were detected by Quanti PicoGreen assay (Sigma-Aldrich) and HEK-Blue LPS Detection Kit 2 (InvivoGen™, San Diego, CA, USA), while ELISA (Invitrogen, Carlsbad, CA, USA) was used for detection of cytokines (TNF-α, IL-6, and IL-10). In parallel, blood bacterial abundance (bacteremia) was evaluated using the direct spread of mouse blood onto blood agar plates (Oxoid, Hampshire, UK) in serial dilutions and incubating at 37 °C for 24 h before colony enumeration. For the kidney injury score, the injury score was semi-quantitatively evaluated on hematoxylin and eosin (H&E) staining in 4 mm thick paraffin-embedded slides at 200× magnification by the area of injury (tubular epithelial swelling, loss of brush border, vacuolar degeneration, necrotic tubules, cast formation, and desquamation) using the following score: 0, area < 5%; 1, area 5–10%; 2, area 10–25%; 3, area 25–50%; 4, area > 50% [23]. In parallel, for spleen apoptosis, spleens with 10% formalin fixation were stained by anti-active caspase 3 antibody (Cell Signaling Technology, Beverly, MA, USA), using immunohistochemistry, and expressed in positive cells per high-power field (200× magnification) as previously published [23].

### 4.4. Bone Marrow-Derived Macrophages and the In Vitro Experiments

Bone marrow-derived macrophages from mouse femurs using supplemented Dulbecco’s Modified Eagle’s Medium (DMEM) with conditioned medium of the L929 cells (ATCC CCL-1) were derived as previously described [65,101,102,103]. The macrophages at 5 × 10^4^ cells/well in supplemented DMEM (Thermo Fisher Scientific, Waltham, MA, USA) were incubated in 5% carbon dioxide (CO_2_) at 37 °C for 24 h before being treated by 3 different protocols, including (i) a single LPS stimulation; started with DMEM followed by LPS (100 ng/mL) 24 h later (N/LPS), or (ii) LPS tolerance; using the twice stimulations by 100 ng/mL of LPS (LPS/LPS), or control (N/N) using the twice DMEM incubation, before the sample collection (supernatant and cells). Notably, the supernatant of the stimulated cells in all groups was gently removed and washed with DMEM before the re-administration of LPS or DMEM, as previously mentioned [39,87,104]. Supernatant interleukin (IL)-1β was evaluated by ELISA (Invitrogen, Carlsbad, CA, USA) and the gene expression was evaluated by quantitative real-time polymerase chain reaction (PCR), as previously described [102,105,106,107]. Briefly, the RNA was extracted from the cells with TRIzol Reagent (Invitrogen, Carlsbad, CA, USA) together with RNeasy Mini Kit (Qiagen, Hilden, Germany) as 1 mg of total RNA was used for cDNA synthesis with iScript reverse transcription supermix (Bio-Rad, Hercules, CA, USA). Quantitative real-time PCR was performed on a QuantStudio 6 real-time PCR system (Thermo Fisher Scientific, Waltham, MA, USA) using SsoAdvance Universal SYBR Green Supermix (Bio-Rad, Hercules, CA, USA). Expression values were normalized to beta-actin (*β-actin*) as an endogenous housekeeping gene and the fold change was calculated by the ∆∆Ct method. The primers used in this study are listed in Table 1.

### 4.5. Mass Spectrometry Proteomic and Secretome Analysis

The proteomic and secretome analyses were performed, using the cells and supernatant media, respectively, according to previous publications [35,56,104]. Briefly, for proteome analysis, the activated macrophages (1 × 10^6^ cells/well) with 3 protocols, N/N, N/LPS, and LPS/LPS as mentioned above, were processed for in-solution digestion. For secretome analysis, an equal volume of culture medium from 3 conditions was centrifuged to remove intact cells, concentrated by centrifugation in an Amicon Ultracel—3K (EMD Millipore, Billerica, MA, USA), and the buffer exchanged using 8 M urea lysis buffer. The concentrated proteins were also subjected to in-solution digestion. Then, the peptides from N/N, N/LPS, and LPS/LPS from the cells and culture media samples, for proteome and secretome, respectively, were labeled with light reagents (CH2O and NaBH3CN), medium reagents (CD2O and NaBH3CN), and heavy reagents (13CD2O and NaBD3CN), respectively. The pooled peptides were fractionated using a high pH reversed-phase peptide fractionation kit (Thermo Fisher Scientific, San Jose, CA, USA). Liquid chromatography–tandem mass spectrometry (LC-MS/MS) analysis of samples was performed on an EASY-nLC1000 system coupled to a Q-Exactive Orbitrap Plus mass spectrometer equipped with a nanoelectrospray ion source (Thermo Fisher Scientific, San Jose, CA, USA). The mass spectrometry (MS) raw files were searched against the Mouse Swiss-Prot Database (17,138 proteins, November 2022) with a list of common protein contaminants. The search parameters were set for the following fixed modifications: carbamidomethylation of cysteine (+57.02146 Da), as well as light, medium, and heavy dimethylation of N termini and lysine (+28.031300, +32.056407, and +36.075670 Da) and variable modification: oxidation of methionine (15.99491 Da). The false positive discovery rate of the identified peptides based on Q-values using The Proteome Discoverer decoy database together with the Percolator algorithm was set to 1%. The relative MS signal intensities of dimethyl labeled peptides were quantified and presented as ratios of single LPS/no stimulation and LPS tolerance/no stimulation. Log 2 of the ratios in triplicate was used to calculate the *p*-values using Student’s *t*-test. The proteins with a *p*-value < 0.05 were considered significant proteins, and these proteins were subjected to the online DAVID Bioinformatics Resources 6.8 to investigate the enriched biological processes. The mass spectrometry proteomics data have been deposited to the ProteomeXchange Consortium via the PRIDE partner repository with the dataset identifier PXD041265. Then, the data visualization was performed using R packages. Volcano plots were generated by ggplot2 version 3.4.2. KEGG pathway analyses were generated by PathfindR. Go enrichment analysis was performed using Shiny 0.77 (http://bioinformatics.sdstate.edu/go/) accessed on 20 March 2023.

### 4.6. Statistical Analysis

The results are shown as mean ± S.E.M. All data were analyzed with GraphPad Prism6. Student’s *t*-test or one-way analysis of variance [41] with Tukey’s comparison test was used for the analysis of experiments with two and more than two groups, respectively. The survival analysis was determined by the log-rank test. For all datasets, a *p*-value less than 0.05 was considered significant.

## 5. Conclusions

The Ezh2-deleted macrophages induced fewer activities (proteomic and secretome analyses) after LPS stimulation compared with the control states, supporting the less severe sepsis in *Ezh2* null *(*Ezh*^fl/fl^; LysM-Cre^cre/−^)* over the control *(*Ezh*^fl/fl^; LysM-Cre^−/−^)* mice. The more severe sepsis in CLP after LPS tolerance over CLP alone supported the less effective microbial control during LPS tolerance. The Ezh2 inhibitor was more effective in the CLP model than the CLP after LPS tolerance, perhaps due to the more profound sepsis severity in the latter condition. More studies on the use of Ezh2 blockage in sepsis are warranted.

## Figures and Tables

**Figure 1 ijms-24-08517-f001:**
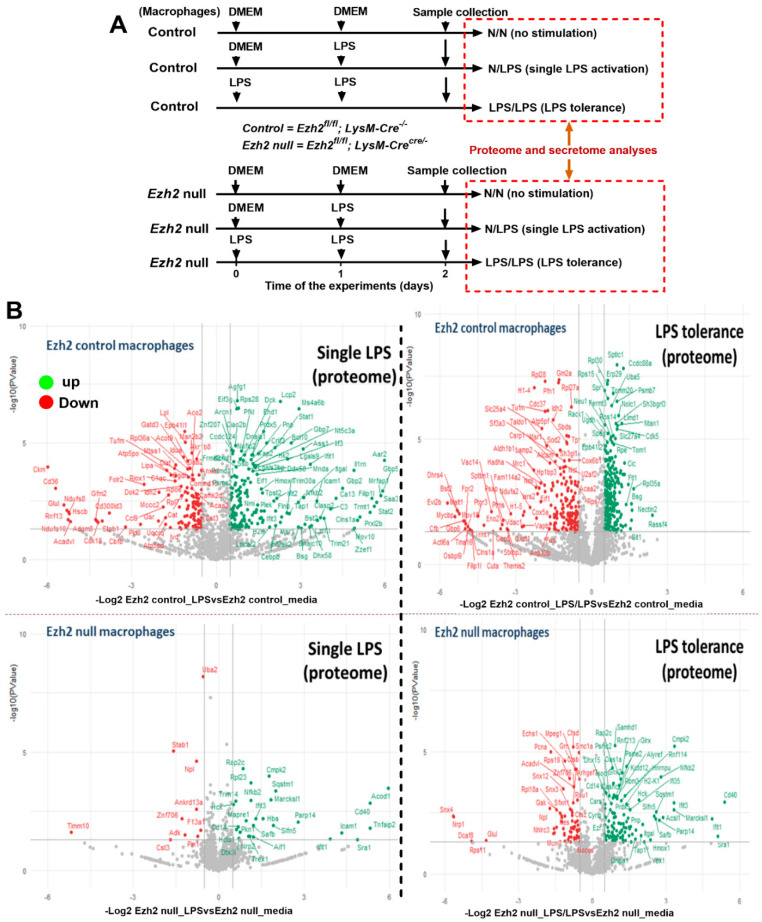
Schema of the experiments using bone marrow-derived macrophages (macrophages) from control mice (Ezh^fl/fl^; LysM-Cre^−/−^) or Ezh2 null mice (Ezh^fl/fl^; LysM-Cre^cre/−^) after activation by lipopolysaccharide (LPS) in a single protocol (N/LPS) that started with the culture media (DMEM) followed by LPS 24 h later or LPS tolerance (LPS/LPS) by the repeated LPS stimulations or no stimulation control (N/N) with DMEM incubation (**A**) is demonstrated. The volcano plot indicating the up- and downregulated proteins, in green and red color, respectively, as compared between the activated cells versus the non-stimulated cells (**B**) is demonstrated. Macrophages were isolated from 3 different mice for the triplicate analysis.

**Figure 2 ijms-24-08517-f002:**
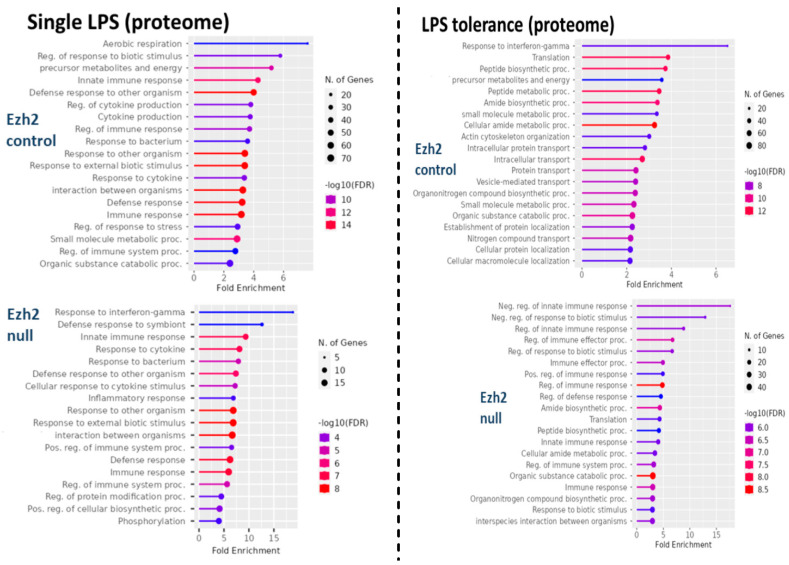
Fold enrichment analysis of cellular proteins in macrophages (proteome) from Ezh2 control (Ezh^fl/fl^; LysM-Cre^−/−^) (Ezh2 control) or Ezh2 null mice (Ezh^fl/fl^; LysM-Cre^cre/−^) (Ezh2 null) after a single lipopolysaccharide (LPS) activation (**left side**) or LPS tolerance (**right side**) between Ezh2 control cells or Ezh2 null macrophages with versus without activation is demonstrated.

**Figure 3 ijms-24-08517-f003:**
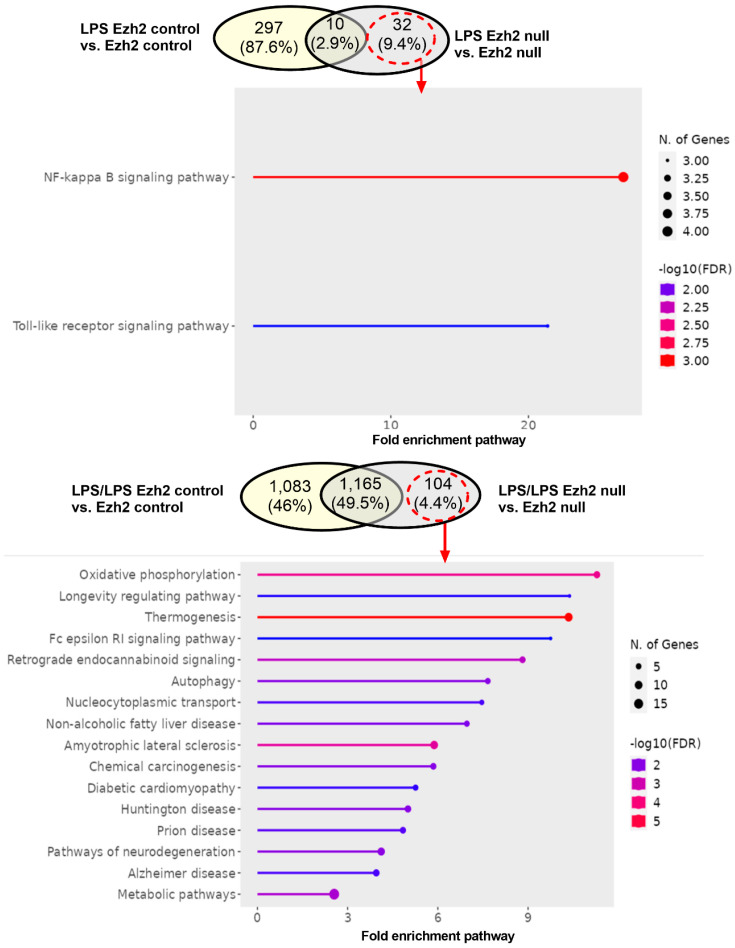
The Venn diagrams of proteome analysis from macrophages of Ezh2 control (Ezh^fl/fl^; LysM-Cre^−/−^) (Ezh2 control) or Ezh2 null mice (Ezh^fl/fl^; LysM-Cre^cre/−^) (Ezh2 null) after a single lipopolysaccharide (LPS) activation (**upper**) or LPS tolerance (**lower**) between Ezh2 control cells or Ezh2 null macrophages versus the non-activated control of each group are demonstrated. The fold enrichment pathway of the unique proteins that presented only in LPS-stimulated macrophages but not in the Ezh2 control cells (dashed circles with arrows) is also demonstrated. Notably, percentages in the Venn diagram are the number of proteins in each part divided by the total number of proteins from both groups.

**Figure 4 ijms-24-08517-f004:**
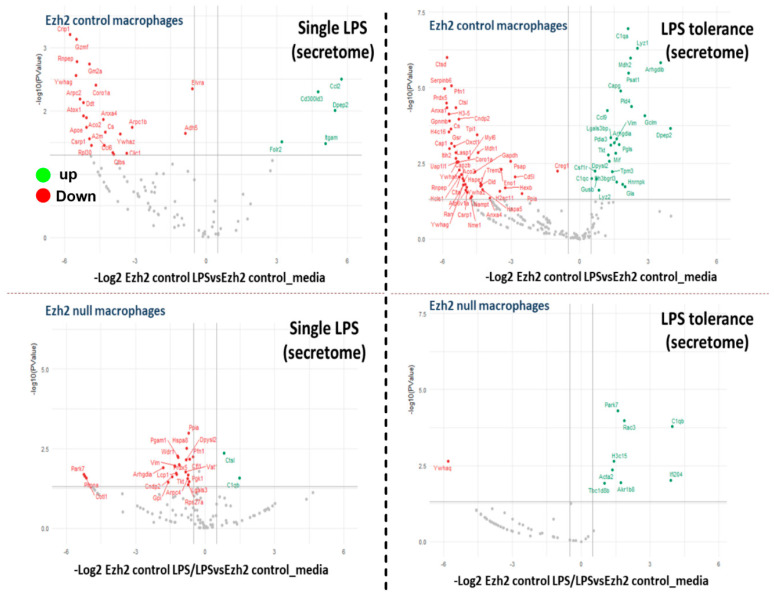
The volcano plots of secretome analysis from macrophages of Ezh2 control (Ezh^fl/fl^; LysM-Cre^−/−^) (Ezh2 control) or Ezh2 null mice (Ezh^fl/fl^; LysM-Cre^cre/−^) (Ezh2 null) after a single lipopolysaccharide (LPS) activation (**left side**) or LPS tolerance (**right side**) between Ezh2 control cells (**upper**) or Ezh2 null macrophages (**lower**) with versus without the activations are demonstrated.

**Figure 5 ijms-24-08517-f005:**
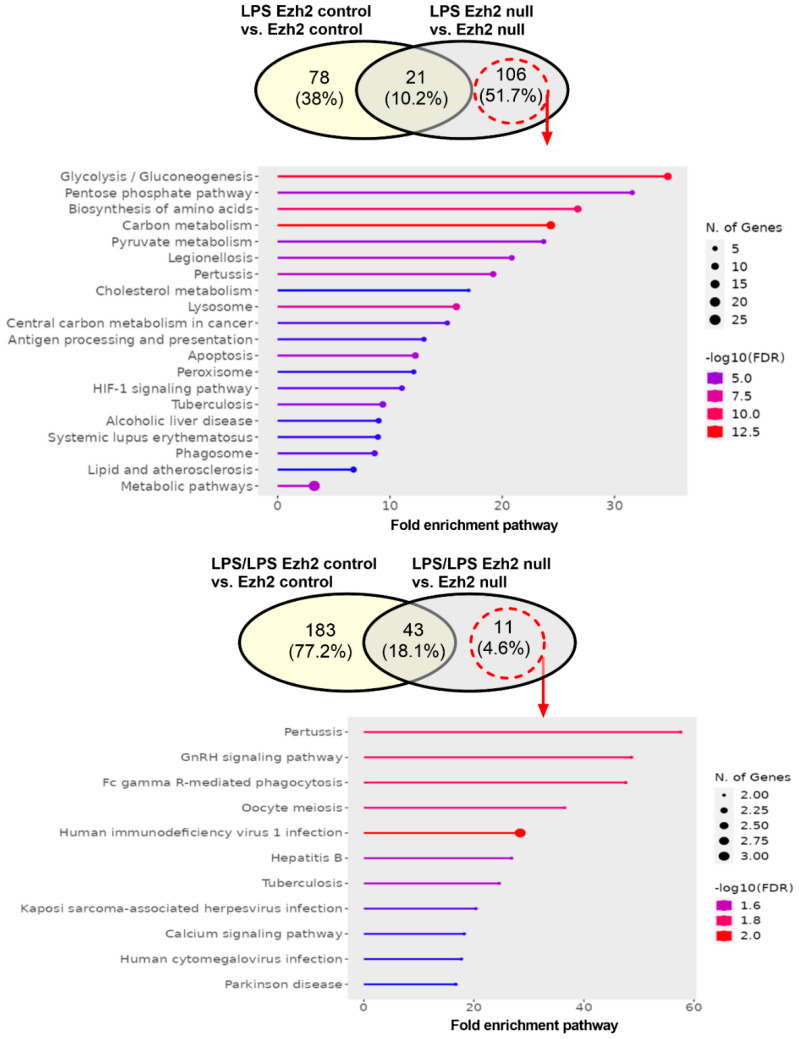
The Venn diagrams of secretome analysis from macrophages of Ezh2 control (Ezh^fl/fl^; LysM-Cre^−/−^) (Ezh2 control) or Ezh2 null mice (Ezh^fl/fl^; LysM-Cre^cre/−^) (Ezh2 null) after a single lipopolysaccharide (LPS) activation (**upper**) or LPS tolerance (**lower**) between Ezh2 control cells or Ezh2 null macrophages versus the non-activated control of each group are demonstrated. The fold enrichment pathway of the unique proteins that presented only in LPS-stimulated macrophages but not in the Ezh2 control cells (dashed circles with arrows) is also demonstrated. Notably, percentages in the Venn diagram are the number of proteins in each part divided by the total number of proteins from both groups.

**Figure 6 ijms-24-08517-f006:**
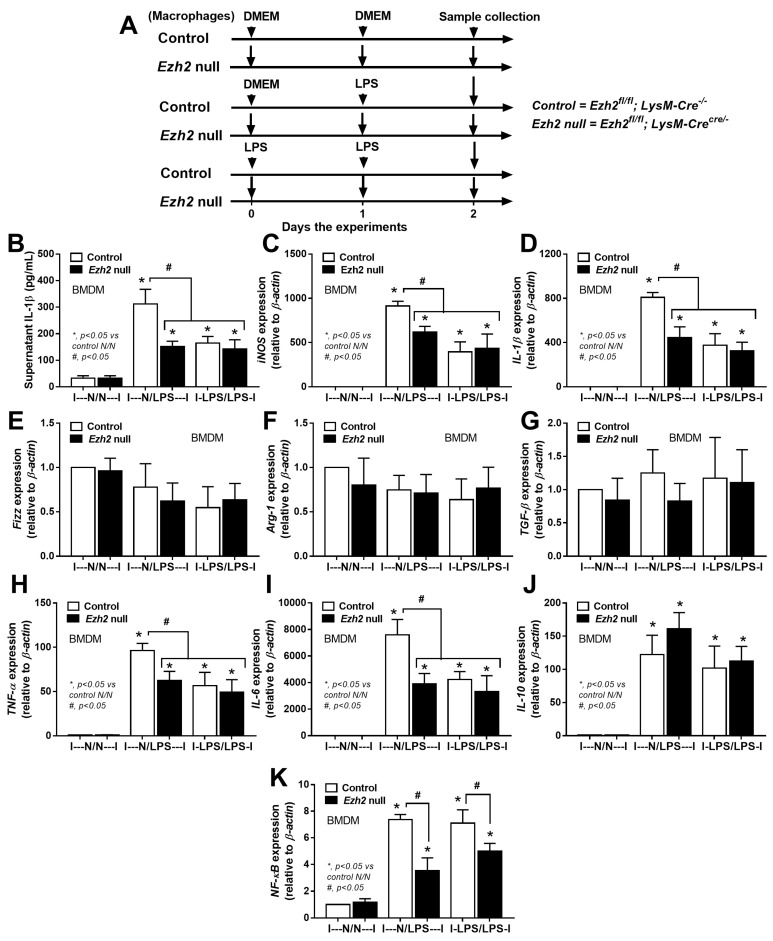
The schema of the experiments in bone marrow-derived macrophages (BMDM) from Ezh2 control (Ezh^fl/fl^; LysM-Cre^−/−^) or Ezh2 null (Ezh^fl/fl^; LysM-Cre^cre/−^) mice after activation by lipopolysaccharide (LPS) in a single protocol (N/LPS) that started with the culture media followed by LPS at 24 h later or LPS tolerance (LPS/LPS) by the twice LPS stimulations or control (N/N) using the twice culture media incubation (**A**) is demonstrated. The characteristics of these macrophages with different protocols, as indicated by supernatant IL-1β (**B**) with the expression of genes for M1 macrophage polarization (*IL-1β* and *iNOS*) and M2 polarization (*Fizz-1, Arg-1,* and *TGF-β*) (**C**–**G**), inflammatory genes (*TNF-α*, *IL-6*, and *IL-10*) (**H**–**J**), and inflammatory mediators (*NF-κB*) (**K**) are demonstrated. Triplicated independent experiments were performed. Mean ± SEM is presented with the one-way ANOVA followed by Tukey’s analysis (*, *p* ˂ 0.05 vs. WT N/N and #, *p* ˂ 0.05).

**Figure 7 ijms-24-08517-f007:**
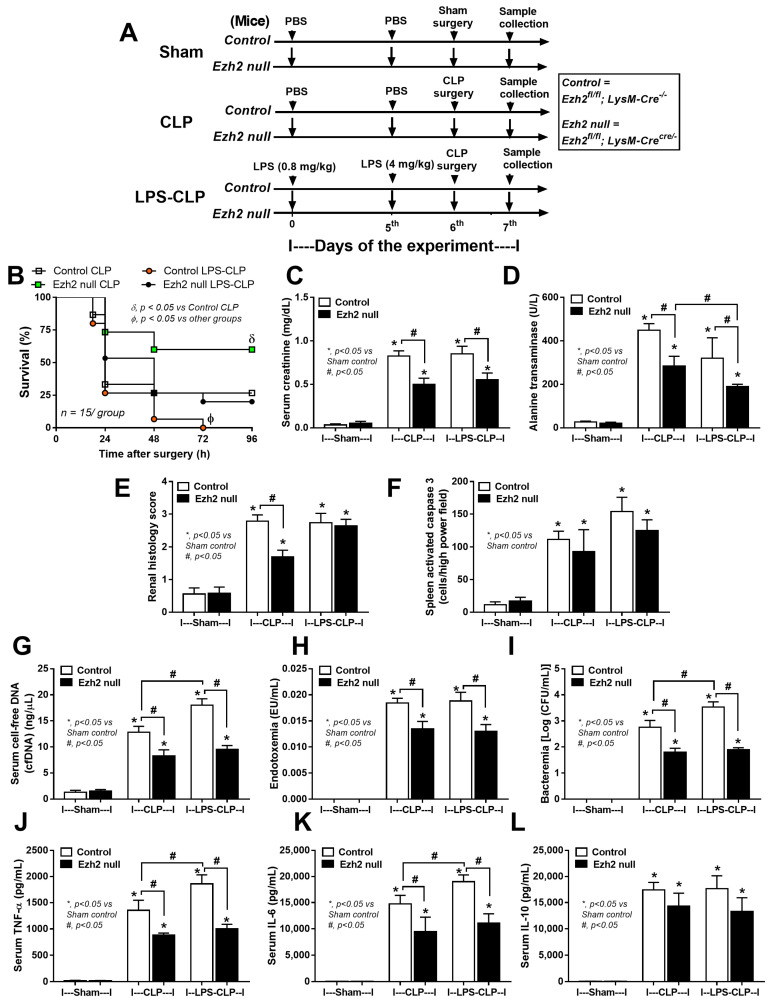
The schema of the experiments in Ezh2 control (Ezh^fl/fl^; LysM-Cre^−/−^) (Control) or Ezh2 null (Ezh^fl/fl^; LysM-Cre^cre/−^) mice (Ezh2 null) in sham, cecal ligation and puncture surgery (CLP), and lipopolysaccharide (LPS) tolerance before CLP surgery (LPS-CLP). The protocol start by injection of phosphate buffer solution (PBS) or LPS intraperitoneal (ip) injection (0.8 mg/kg) followed by PBS or LPS (ip 4 mg/kg) at 5th day before sham or CLP surgery at 6th day, and sacrifice with sample collection at 7th day of the experiment (**A**). Characteristics of these mice as indicated by survival analysis (**B**), kidney injury (serum creatinine) (**C**), liver damage (alanine transaminase) (**D**), renal injury score (**E**), spleen apoptosis (**F**), cell-free DNA (**G**), endotoxemia (**H**), bacteremia (**I**), and serum cytokines (TNF-α, IL-6, and IL-10) (**J**–**L**) are demonstrated (n = 15/group for B and n = 5–7/group for (**C**–**J**)). Mean ± SEM is presented with the one-way ANOVA followed by Tukey’s analysis (*, *p* ˂ 0.05 vs. Sham control and #, *p* ˂ 0.05).

**Figure 8 ijms-24-08517-f008:**
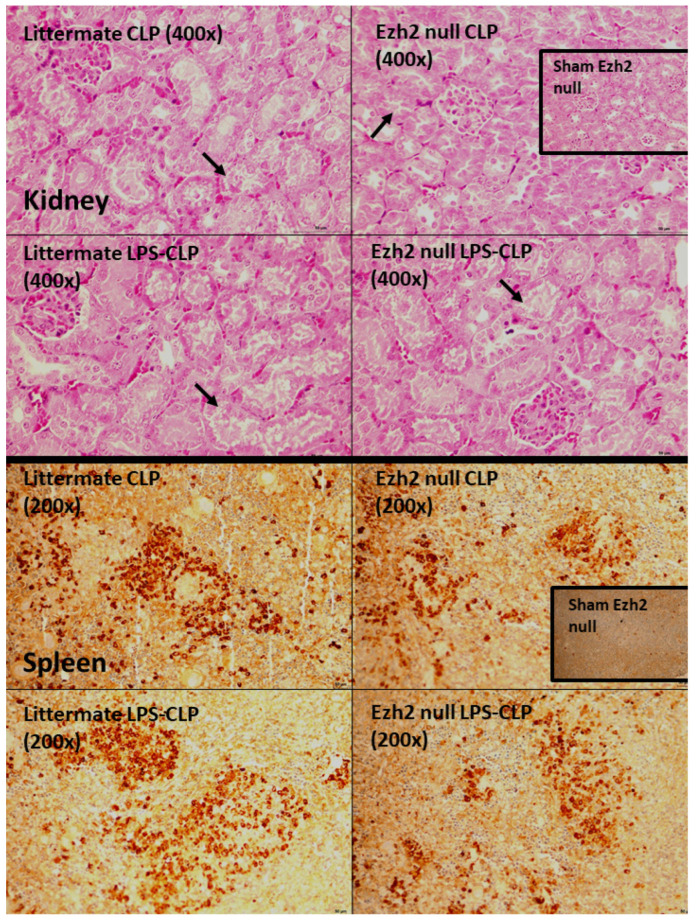
Representative pictures of renal histology with hematoxylin and eosin (H&E) stain (**upper**) and activated caspase 3 spleen apoptosis (**lower**) of Ezh2 control (Ezh^fl/fl^; LysM-Cre^−/−^) (littermate) or Ezh2 null (Ezh^fl/fl^; LysM-Cre^cre/−^) mice (Ezh2 null) in cecal ligation and puncture surgery (CLP), and lipopolysaccharide (LPS) tolerance before CLP surgery (LPS-CLP) are shown. Only the sham of Ezh2 null mice, but not the sham of littermate control, in renal histology and spleen apoptosis are demonstrated in the inset pictures due to the non-difference between both shams. Arrows indicate an example of renal tubular cell injury.

**Figure 9 ijms-24-08517-f009:**
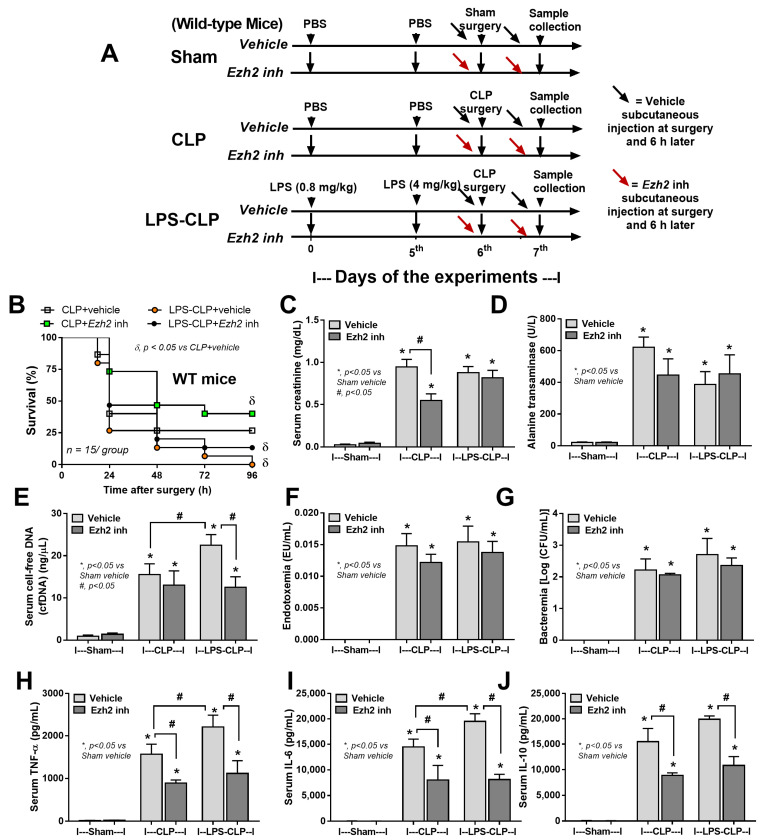
The schema of the experiments in wild-type mice in sham, cecal ligation and puncture surgery (CLP), and lipopolysaccharide (LPS) tolerance before CLP surgery (LPS-CLP). The protocol start by injection of phosphate buffer solution (PBS) or LPS intraperitoneal (ip) injection (0.8 mg/kg) followed by PBS or LPS (ip 4 mg/kg) at 5th day before sham or CLP surgery at 6th day, and sacrifice with sample collection at 7th day of the experiment. Mice were subcutaneously administered with vehicle or an Ezh2 inhibitor at 0 and 6 h post-surgery on the 6th day of protocol (**A**). Characteristics of these mice as indicated by survival analysis (**B**), kidney injury (serum creatinine) (**C**), liver damage (alanine transaminase) (**D**), cell-free DNA (**E**), endotoxemia (**F**), bacteremia (**G**), and serum cytokines (TNF-α, IL-6, and IL-10) (**H**–**J**) are demonstrated (n = 15/group for B and n = 5–7/group for (**C**–**J**)). Mean ± SEM is presented with the one-way ANOVA followed by Tukey’s analysis (*, *p* ˂ 0.05 vs. Sham vehicle control and #, *p* ˂ 0.05).

**Table 1 ijms-24-08517-t001:** List of primers used in the study.

Name	Forward	Reverse
Inducible nitric oxide synthase (*iNOS*); Gene ID: 18126	5′-ACCCACATCTGGCAGAATGAG-3′	5′-AGCCATGACCTTTCGCATTAG-3′
Interleukin-1β (*IL-1β*) Gene ID: 16176	5′-GAAATGCCACCTTTTGACAGTG-3′	5′-TGGATGCTCTCATCAGGACAG-3′
Tumor necrosis factor α (*TNF-α*) Gene ID: 21926	5′-CCTCACACTCAGATCATCTTCTC-3′	5′-AGATCCATGCCGTTGGCCAG-3′
Interleukin-6 *(IL-6)* Gene ID: 16193	5′-TACCACTTCACAAGTCGGAGGC-3′	5′-CTGCAAGTGCA TCA TCGTTGTTC-3′
Interleukin-10 (*IL-10*) Gene ID: 16153	5′-GCTCTTACTGACTGGCATGAG-3′	5′-CGCAGCTCTAGGAGCATGTG-3′
Arginase-1 (*Arg-1*) Gene ID: 11846	5′-CTTGGCTTGCTTCGGAACTC-3′	5′-GGAGAAGGCGTTTGCTTAGTT-3′
Resistin-like molecule-α1 (*FIZZ-1*) Gene ID: 57262	5′-GCCAGGTCCTGGAACCTTTC-3′	5′-GGAGCAGGGAGATGCAGATGA-3′
Transforming growth factor-β (*TGF-β*) Gene ID: 21813	5′-CAGAGCTGCGCTTGCAGAG-3′	5′-GTCAGCAGCCGGTTACCAAG-3′
Nuclear factor kappa B (*NFκB*) Gene ID: 18033	5′-CTTCCTCAGCCATGGTACCTCT-3′	5′-CAAGTCTTCATCAGCATCAAACTG-3′
*β-actin* Gene ID: 11461	5′-CGGTTCCGATGCCCTGAGGCTCTT-3′	5′-CGTCACACTTCATGATGGAATTGA-3′

## Data Availability

Not applicable.

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
