# Peer review of "Less Severe Sepsis in Cecal Ligation and Puncture Models with and without Lipopolysaccharide in Mice with Conditional Ezh2-Deleted Macrophages (LysM-Cre System)"

_ijms, 2023, doi:10.3390/ijms24108517_

Round 1

Reviewer 1 Report

See attachment.

Author Response

Reviewer 1

While the results of this study are promising, there are limitations that need to be addressed:

1) There are sex-specific differences in sepsis as well as in the effects of therapeutic treatments.

However, only male mice were used in the mouse model. The authors should justify their

choice of using only male mice and discuss any potential limitations resulting from this

decision.

ANS: We thank the reviewer for the comment and put it as a limitation of the study in the new discussion as following “Only male mice were used and the impact of gender differences in sepsis might affect the translation of the results.”.

2) The use of a mouse model allows for the examination of organ tissues, such as the kidney, liver,

and heart. The study would benefit from histological analysis of these organs to assess tissue

inflammation and further support the findings on sepsis severity. The authors should consider

incorporating these analyses into their study.

ANS: We thank the reviewer for the comment and put renal and spleen histology into the results.

3) The methods section lacks a description of the mortality assay and the primer sequences for

NF-κB. Please provide this information.

ANS: We apologize for the mistake and add the information accordingly (Log rank test and NF-kB primer sequence).

4) The study analyzes NF-κB and iNOS at the mRNA level using qRTPCR, which is unusual and

provides limited information about their activation status. The authors should consider using

Western blots for these targets (at least in BMDMs) to gain more meaningful insights into their

translocation and function.

ANS: We thank the reviewer for the comment and agree that this will increase the value of our manuscript. However, we have an immediate problem on Ezh2 null mice and the frozen Ezh2 null cells. Then, we put this as a limitation of our work in the discussion as following “Second, details of mechanistic pathways, including the Western blot analysis, were not performed. Despite a proof of concept for the translational use of Ezh2 inhibitors in clinical sepsis, more experiments are interesting.”.  

5) Figures 2 and 4 should be made clearer and more organized for better understanding.

ANS: We thank the reviewer for the comment and organize and dividing into more figures.

6) In line 304, the term "1 macrophage polarization" should be corrected to "M1 macrophage

polarization."

ANS: We thank the reviewer for the comment and correct it accordingly.

7) In line 334, the reference to Figure 5A should be corrected to Figure 5B. The authors should

review and verify all cross-references throughout the manuscript.

ANS: We apologize for the mistake and correct it accordingly.

8) The Figures 6 and 7 display an inconsistency between the sample sizes mentioned in the figure

(n=15) and the figure caption (n=1). The authors should correct this discrepancy.

ANS: We thank the reviewer for the comment and correct it accordingly.

9) I may be wrong, but is it possible that the figure 5K is already published in the following

publication? Kunanopparat, A.; Leelahavanichkul, A.; Visitchanakun, P.; Kueanjinda, P.;

Phuengmaung, P.; Sae-khow, K.; Boonmee, A.; Benjaskulluecha, S.; Palaga, T.; Hirankarn, N. The

Regulatory Roles of Ezh2 in Response to Lipopolysaccharide (LPS) in Macrophages and Mice

with Conditional Ezh2 Deletion with LysM-Cre System. Int. J. Mol. Sci. 2023, 24, 5363.

https://doi.org/10.3390/ijms2406536

ANS: We thank the reviewer for the comment. This is a repeated experiment; however, we agree that this is the same information for the communication. Then, we cut fig 5K (Socs3) and just refer it from the previous publication in the new version of the manuscript.

Reviewer 2 Report

The manuscript is well written and the experimental design is well orchestrated to show that the proteome and secretome expression profiles were different in macrophage cells (in Ezh2 conditional knock-out mice) owing to the differential influence of Ezh2 on LPS tolerance. The figures clearly explain the author’s intent and are well representative of the results. A few minor changes when addressed can enhance the quality of the manuscript and make it more suitable for publishing in IJMS journal.

Please find the comments below:

Page 4: Typo: line 202: letter, should be "latter"

Page 10: Lines 294-299: the relation between histone 3 lysine 27 methylation and ezh2 in macrophages is unclear, and it is not clear why the authors mentioned it.

Page 15: lines 534: “In contrast, the Ezh2 suppression can also enhanced pro535 inflammatory genes, such as NF-κB, that worsens inflammatory bowel 536 diseases and sepsis-induced muscle cell apoptosis [49,74-76].” Typo in this sentence.

Better conclusion could have been written. A more comprehensive conclusion can be written including the concrete results. it would help the authors to re-write or rephrase the conclusion.

English language is very good and apart from a few typos, I have not had any difficulty understanding the content.

Author Response

Reviewer 2

The manuscript is well written and the experimental design is well orchestrated to show that the proteome and secretome expression profiles were different in macrophage cells (in Ezh2 conditional knock-out mice) owing to the differential influence of Ezh2 on LPS tolerance. The figures clearly explain the author’s intent and are well representative of the results. A few minor changes when addressed can enhance the quality of the manuscript and make it more suitable for publishing in IJMS journal.

Please find the comments below:

Page 4: Typo: line 202: letter, should be "latter"

ANS: We thank the reviewer for the comment and correct it accordingly.

Page 10: Lines 294-299: the relation between histone 3 lysine 27 methylation and ezh2 in macrophages is unclear, and it is not clear why the authors mentioned it.

ANS: We thank the reviewer for the comment and cut this part.

Page 15: lines 534: “In contrast, the Ezh2 suppression can also enhanced pro535 inflammatory genes, such as NF-κB, that worsens inflammatory bowel 536 diseases and sepsis-induced muscle cell apoptosis [49,74-76].” Typo in this sentence.

ANS: We thank the reviewer for the comment and correct it into “In contrast, suppression of Ezh2 might enhance pro-inflammatory genes, including NF-κB, that possibly worsens inflammatory bowel diseases and muscle cell apoptosis in sepsis”.

Better conclusion could have been written. A more comprehensive conclusion can be written including the concrete results. it would help the authors to re-write or rephrase the conclusion.

ANS: We thank the reviewer for the comment and correct it accordingly.

Reviewer 3 Report

It is well documented that EZh2 plays roles in sepsis although the authors have not fully cited these papers. This manuscript mainly focused on ezh2 deficient macrophages with LPS treatment/tolerance. They presented the difference between parental control and ezh2 deficiency in macrophages, and concluded that inhibition of EZH2 function reduced sepsis severity, similar to previous reports. They proposed that Soc3 may be involved in this process.

Minor points:

The x-labels in bar charts seem a bit odd. Also in figure 6A and 7A.

The primers shall have Genebank ID and start /end numbers

Statistical analysis: shall include the methods for comparing survival rates

Discussion of the limitation of this work

Some paragraphs in discussion seem too long, better no subtitles in discussion

The Figure 8 is too subjective and no links between input and output. It seems to make more confusing. Better without it.

none

Author Response

Reviewer 3

It is well documented that EZh2 plays roles in sepsis although the authors have not fully cited these papers. This manuscript mainly focused on ezh2 deficient macrophages with LPS treatment/tolerance. They presented the difference between parental control and ezh2 deficiency in macrophages, and concluded that inhibition of EZH2 function reduced sepsis severity, similar to previous reports. They proposed that Soc3 may be involved in this process.

Minor points:

The x-labels in bar charts seem a bit odd. Also in figure 6A and 7A.

ANS: We thank the reviewer for the comment and correct it into “Days of experiments”.

The primers shall have Genebank ID and start /end numbers

ANS: We thank the reviewer for the comment and correct it accordingly.

Statistical analysis: shall include the methods for comparing survival rates

ANS: We thank the reviewer for the comment and correct it accordingly.

Discussion of the limitation of this work

ANS: We thank the reviewer for the comment and correct it accordingly.

Some paragraphs in discussion seem too long, better no subtitles in discussion

ANS: We thank the reviewer for the comment and correct it accordingly.

The Figure 8 is too subjective and no links between input and output. It seems to make more confusing. Better without it.

ANS: We thank the reviewer for the comment and cut this part.

Round 2

Reviewer 1 Report

Dear authors, 

Thank you for answering the questions and revising the manuscript.